# Designing New Sustainable Polyurethane Adhesives: Influence of the Nature and Content of Diels–Alder Adducts on Their Thermoreversible Behavior

**DOI:** 10.3390/polym14163402

**Published:** 2022-08-19

**Authors:** Susana Quiles-Díaz, Helga Seyler, Gary J. Ellis, Peter S. Shuttleworth, Araceli Flores, Marián A. Gómez-Fatou, Horacio J. Salavagione

**Affiliations:** Departamento de Física de Polímeros, Elastómeros y Aplicaciones Energéticas, Instituto de Ciencia y Tecnología de Polímeros (ICTP), Consejo Superior de Investigaciones Científicas (CSIC), c/Juan de la Cierva 3, 28006 Madrid, Spain

**Keywords:** reusable adhesive, high reversibility, solvent-based, solvent-free

## Abstract

With a view to the development of new sustainable and functional adhesives, two Diels–Alder (DA) adducts are incorporated as a third component into the curing process of solvent-based and solvent-free polyurethanes in this study. The influence of the nature and content of the DA molecules on the retro-DA (rDA) reaction and its reversibility and cyclability is investigated. It is demonstrated that the bonding/debonding properties of the adhesives are mainly controlled by the concentration of the DA adducts, with a minimum thermoreversible bond (TB) content required that depends on the system and the total ratio between all the diols in the formulation. For the solvent-based system, rDA/DA reversibility can be repeated up to ~20 times without deterioration, in contrast to the solvent-free system where a gradual loss in the DA network reconstruction efficiency is observed. Despite this limitation, the solvent-free system presents clear advantages from an environmental point of view. The changes observed in the physical properties of these new thermoreversible adhesives are of great relevance for recycling strategies and, in particular, their potential for separating multilayered film packaging materials in order to recycle the individual polymer films involved.

## 1. Introduction

Packaging accounts for around 40% of the plastics market, but is responsible for over 60% of post-consumer plastic waste in the EU, of which only 40% is recycled [1]. This is a major problem and the search for strategies to increase recycling in this area is of paramount importance for a more sustainable future. However, the complexity of packaging materials is continuously increasing due to the demand for multiple functions. Thus, multilayered materials are common, especially for food packaging, since they can combine different polymer films with diverse functionalities, providing excellent properties with respect to food health and safety and increased shelf life. However, their impact is very negative from an environmental point of view since these materials are very difficult to recycle and usually end up incinerated or landfilled. The main reason behind this lies in the lack of effective methods for the separation of different polymer layers into clean waste streams that prevents the reuse of each of the component materials [2].

Currently, the routes proposed to solve the recycling problems of multilayer systems can be divided into three main approaches: (i) the development of innovative methods to separate the different components, (ii) the combined processing of all the constituents, which limits to some extent the range of alternatives for posterior applications, and (iii) the ambition to find a monolayer and/or monomaterial solutions to substitute the multilayer functions [2,3,4]. Focusing on the first approach, one promising possibility is to act on the adhesive layer that binds the polymer layers together. The work described here is based on this strategy and the main objective is to develop new functional adhesives with reversible adhesion/separation capabilities in order to facilitate the recycling of multilayered packaging.

Dynamic covalent bonds have been previously used mainly for polymer self-healing but also in recent years for recycling [4]. Amongst these, the Diels−Alder (DA) reaction has been employed in the design of self-healing polymers, especially polyurethanes (PU) [5,6,7,8,9,10]. Lately, it has also been used for recycling [11,12,13]. These studies have demonstrated the possibility of incorporating DA bonds that can be switched on and off with temperature, providing a very versatile mechanism for incorporating new functionalities in PU.

Here, we report on the influence of incorporating different DA adducts into two types of commercial polyurethane adhesives (solvent-based and solvent-free adhesives), particularly focusing on the cyclability of the adhesives for the recycling of polymer laminates. The presence of dynamic bonds confers to the adhesives a new functionality, i.e., the possibility to lose adhesion via a simple thermal treatment. This means that the adhesive behaves as such and can be used for sticking polymer layers together in polymer laminates and after use it can be modified by heating, which leads to a loss of adhesion, facilitating the recycling of individual polymer layers. Different adhesive systems are used depending on the polymer films to be bonded, for instance, solvent-based adhesives are selected for binding polyesters, while solvent-free adhesives are preferred for polylolefins. One of the main challenges of this investigation has been to incorporate these adducts as a third component during the curing process of the adhesives with only minor modifications to the standard industrial process in order to facilitate scale-up and extrapolation to real commercial applications. The influence of the chemical structure and content of the DA adducts, the feed ratio between the different adhesive components, and the adhesives type are all analyzed in order to understand their thermoreversible behavior and its impact on the physical properties of the prepared adhesives. In the particular case of solvent-based adhesives, it is demonstrated that the process, i.e., the rDA/DA cycle, can be repeated up to ~20 times for both adducts with no significant loss in the enthalpy of the process. This is a very promising result for reversible adhesives, and it is expected to greatly impact on the recycling of multilayer films for packaging.

## 2. Materials and Methods

### 2.1. Materials

The DA adducts were synthesized by reaction with furfuryl alcohol (FA) (Sigma-Aldrich, Darmstadt, Germany. CAS number: 98-00-0, 98%), which was distilled before use, with two different bismaleimides (BMI): 1,1′-(methylenedi-4,1-phenylene)bismaleimide (BMIF) (Sigma-Aldrich, CAS number: 13676-54-5) and N,N′-(1,3-phenylene)dimaleimide (BMIR) (Sigma-Aldrich, CAS number: 3006-93-7, 97%). The isolation of the DA adducts was undertaken by precipitation in diethyl ether containing BHT as an inhibitor (DEE, Sigma-Aldrich, CAS number: 60-29-7), which was used as received.

1,4-Dioxane (Panreac, Castellar del Vallès, Barcelona, Spain, CAS number: 123-91-1) was used to assess the evolution of the rDA reaction as a function of temperature for the DA adduct molecules in solution.

The adhesive systems employed, composed of commercial isocyanate prepolymers and polyol catalysts (a mixture of hexanedioic acid, polymer with 2,2-dimethyl-1,3-propanediol and 1,6-hexanediol), were kindly provided by SAPICI SpA, Milan, Italy). Ethyl acetate (EA, Honeywell, Charlotte, NC, USA, CAS number: 141-78-6) was used as eluent in the solvent-based adhesive preparation.

### 2.2. Synthesis of the DA Adducts

The synthesis of the DA adduct with BMIR was conducted as follows: 5 g of BMI (18.6 mmol) were dissolved in 22.5 mL of FA (260 mmol). The solution was maintained at 55 °C for 21 h under reflux for the DA reaction to proceed. The reaction product was precipitated dropwise in 500 mL of DEE under vigorous stirring. The DA adduct was then vacuum-filtered using Whatman^®^ 42 double-filter paper. The reaction product was thoroughly washed with DEE (300 mL) at least twice, followed by vacuum filtration as stated in the previous step. Finally, the isolated product was dried at 40 °C under vacuum until constant weight was reached. 

For the synthesis of the DA adduct with BMIF, 5.4 g of BMI (14.3 mmol) was dissolved in 8 mL of FA (92.6 mmol). The solution was stirred and maintained at 40 °C for 24 h under reflux for the DA reaction to proceed. The reaction product was precipitated dropwise in 500 mL of DEE at 35 °C under vigorous stirring. The DA adduct was then vacuum-filtered using Whatman^®^ 42 double-filter paper. The reaction product was thoroughly washed with DEE at 35 °C (300 mL) at least twice, followed by vacuum filtration. Finally, the isolated product was dried at 40 °C under vacuum until constant weight was reached.

### 2.3. Adhesive Formulations Containing DA Adducts

For the preparation of the solvent-based adhesive, the required amounts of polyol catalyst, DA molecules, FA (if used), EA and isocyanate prepolymer were used. Depending on the target amount of thermoreversible bonds (TB) the amounts of components employed ranged from 100 to 200 mg for polyol, 12.9 to 31.9 mg for DAR, 15.4 to 37.0 mg for DAF and 2.6 to 6.6 mg for FA (when used). The amount of isocyanate prepolymer remained constant at 1 g for all adhesive formulations. In a typical experiment, all components were placed in a polypropylene container and homogenized using a Thinky ARE-250 planetary centrifugal mixer (PCM) (Thinky Corporation, Tokyo, Japan) applying the following program: 60 s at 500 rpm, 20 s at 2000 rpm, 60 s at 2200 rpm (degassing), and 30 s at 1000 rpm. An aliquot of the adhesive was then cured at 50 °C for 20 h and then left at RT for a further 6 days in a polypropylene mold. The formulations were prepared with an NCO/OH mole ratio index of 1.5.

The preparation procedure of the solvent-free adhesive was carried out in two steps. The dispersion of the adduct in this system is a great challenge and requires prolonged mixing times. First, the mixture of the required amounts of polyol catalyst (480–800 mg), DAF (86.1–172.2 mg) and FA (31.2–52 mg, if used) was subjected to a PCM treatment at room temperature with the following program: 9 min at 2000 rpm and 9 min at 2200 rpm (degassing step). Secondly, the required amount of isocyanate prepolymer (1 g) was added to the mixture and was further homogenized at room temperature with the application of the following program: 6 min at 2000 rpm and 6 min at 2200 rpm (degassing). The adhesives were initially cured at 50 °C for 20 h and then left at RT for the following 6 days. The formulations were prepared with an NCO/OH mole ratio index of 1.3.

### 2.4. Characterization

The DA adducts were dissolved in deuterated acetone and then characterized using ^1^H NMR and ^13^C NMR on a Bruker Advance III-400 MHz (Bruker Scientific Instruments, Billerica, MA, USA). MestReNova software was used for data processing and the analysis of the NMR spectra. UV–Visible spectra of the samples were recorded in quartz cuvettes with a Perkin Elmer Lambda 35 spectrometer (Perkin Elmer Inc., Waltham, MA, USA) over the spectral range from 190 to 450 nm. Experiments were conducted in solution, with 50 mg of the DA adducts dissolved in 100 mL of 1,4-dioxane. The solutions were then heated to 70 and 90 °C and maintained at these temperatures for different time intervals to monitor the evolution of the rDA reaction by UV–Visible spectroscopy.

The thermal stability of all adhesives was investigated by TGA using a TA Instrument Q50 thermobalance (New Castle, DE, USA) in the range of 50−800 °C at a heating rate of 10 °C min^−1^ under an inert atmosphere (nitrogen, 60 cm^3^ min^−1^).

Mechanical properties were assessed by dynamic depth-sensing indentation. Samples of the adhesive films were glued onto a metal holder that was placed on the platform of a G200 nanoindenter (KLA Tencor, Ann Arbor, MI, USA). A diamond flat-end rod with an effective diameter of 49.9 μm was used (assuming a cylindrical shape). In a typical experiment, the indenter penetrates the sample surface up to a precompression distance of 10 μm. After the dissipation of transient creep, the indenter vibrates at a frequency of 1 Hz and the dynamic contact stiffness can be determined from the phase angle between the harmonic indenter displacement and the excitation force. Finally, storage modulus values can be calculated assuming elastic–viscoelastic correspondence.

The rDA reaction in the adhesives was investigated by Fourier transform infrared spectroscopy using a Perkin−Elmer Spectrum Two FTIR spectrometer (Perkin Elmer Inc., Waltham, MA, USA) incorporating a universal attenuated total reflectance (ATR) accessory with a diamond crystal. Spectra were recorded, accumulating 12 scans at 4 cm^−1^ spectral resolution. The rDA conversion was monitored following the signal intensity of the absorbance band corresponding to the maleimide =C−H out of the plane bending vibration that appeared at 690 cm^−1^ or 700 cm^−1^ for BMIF [14] or BMIR, respectively. Films were processed according to the following procedures: (a) they were heated at 120 °C for 15 min and the FTIR spectrum was recorded (first rDA reaction expected); (b) the same film was then isothermally treated at 60 °C for 20 h (network reconstruction via the DA reaction was expected); and (c) films were heated again at 120 °C for 15 min and the FTIR spectrum was subsequently recorded (second rDA reaction expected).

Additionally, differential scanning calorimetry (DSC) was employed to characterize the rDA reaction in the adhesives using a TA Instrument DSC25 with an RSC90 refrigerated cooling system (New Castle, DE, USA). The experiments were carried out under nitrogen atmosphere on samples of approximately 5 mg in hermetically sealed aluminum pans under a nitrogen flow of 50 mL/min. A heating cycle from −40°C to 160 °C at a heating rate of 10 °C /min was applied. Moreover, DSC was also used to evaluate the cyclability and reversibility of the DA and rDA reactions, and the thermal treatment applied consisted of: (i) an initial heating from −40 °C to 160 °C at 10 °C/min (first rDA reaction by DSC) followed by (ii) cooling down to 40 °C at a high rate (40 °C/min) and (iii) isothermal treatment at 60 °C for 20 h (to promote the DA network reconstruction) in an oven. These steps were then repeated 20 times. In the case of the solvent-free adhesives, the same thermal treatments were used but over a wider temperature range (from −70 °C to 200 °C) and, for this system, only 10 cycles were applied.

To analyze the influence of temperature on the physical properties of the adhesives, films were cut into small pieces that were stacked one on top of another. These stacked pieces were then subjected to a thermal cycle that consisted of heating at 120 °C for 15 min under the application of a slight pressure (using a 450 g or a 900 g brass weight for the solvent-based and solvent-free systems, respectively). The sample was then cooled to 60 °C and maintained at this temperature for at least 20 h.

## 3. Results and Discussion

### 3.1. Preparation of the Thermoreversible DA Adducts

As previously described in the introduction, one of the objectives of this work was to synthesize specific molecules based on diols containing DA bonds to incorporate as a third component to the diol/isocyanate PUR precursors in order to introduce thermoreversible bonds along the adhesive PUR network.

The scheme of the DA reaction and the two bismaleimide molecules employed are shown in Figure 1. As can be seen, both bismaleimide molecules are symmetrical. The structural difference between these molecules lies in the chemical groups connecting both sides. For 1,1′-(methylene-di-1,4-phenylene)bismaleimide (BMIF), the methylene group confers to the molecule a degree of rotation around this axis, making this molecule more “flexible”. In the case of *N,N′-*(1,3-phenylene)bismaleimide (BMIR), the molecule is more rigid due to the presence of a single aromatic ring between both sides containing the functional alkene group. Thus, the resulting DA adducts are named **DAF** and **DAR**, respectively, due to the relative flexibility or rigidity of the BMI. These differences between both molecules are relevant because they directly affect the properties of the adhesives, as will be shown below.

Several experimental conditions, especially with the use of different organic solvents as reaction media, and different FA:BMI ratios and temperatures were investigated in order to optimize the synthetic procedure. Eventually, FA was selected to be used as both solvent and reactant and it was used in excess with the aim of improving the reaction yield and to assure that all of the BMI in the feed was consumed. The optimum experimental conditions for the synthesis of **DAF** and **DAR** adducts are shown in Table 1.

It can be highlighted that **DAF** synthesis required lower temperatures than **DAR** synthesis. Lower reaction yields were observed for **DAF** due to its higher solubility when compared with **DAR**, hindering its isolation by precipitation. In order to verify the formation of the desired DA molecules, structural characterization of the products was carried out by ^1^H NMR and ^13^C NMR. The ^1^H NMR and ^13^C NMR spectra of BMIF and **DAF** are displayed in Appendix A. The signals observed in both spectra can be perfectly assigned to the chemical structure of **DAF** and are in good agreement with the data in the literature [15,16,17,18,19]. Moreover, the samples were free of unreacted BMI but bands corresponding to FA were clearly observed as expected due to the excess of FA in the feed. A detailed description of the NMR results is provided in the Electronic Supporting Information (ESI).

The thermoreversible behavior of the DA adducts prior to incorporation into the PUR adhesives was investigated using UV–Vis spectroscopy and is discussed in the ESI. The results compiled in Appendix A suggest that, at a typical rDA reaction temperature (90 °C), the **DAR** adduct is more sensitive than **DAF** and a higher degree of rDA conversion is obtained.

### 3.2. Incorporation of Thermoreversible Molecules into Solvent-Based PUR Adhesives

The synthesized DA adducts were incorporated into solvent-based polyurethane adhesives by curing commercial isocyanates and polyols. These DA-containing diol molecules compete with the diols in the reaction with the isocyanate. By careful control of the concentration of DA bonds in the final PUR, it is possible to modulate the bonding/debonding properties of the adhesives by regulating the feed composition (especially the ratio between different diols), the type of DA molecules, and curing time, among other experimental parameters.

In order to optimize the incorporation of the DA molecules, these were dissolved in ethyl acetate together with the solvent-based NCO-terminated and OH-terminated components following the procedure described in the experimental part. Table 2 includes all the solvent-based adhesives prepared in this work. Several feed compositions varying the DA/polyol ratio were formulated to prepare samples with different DA thermoreversible bond contents. In some cases, FA was added to the formulation to increase the adduct solubility, which altered the ratio of the OH groups. Consequently, to maintain the NCO/OH index, lower amounts of DA were used, leading to a decrease in the TB content, as can be observed in Table 2. TB in this study refers to ratio between equivalents of reversible moieties with respect to equivalents of isocyanate in the feed, assuming that all DA molecules have reacted, and is estimated as detailed in the ESI.

The TB values were calculated from Equation (1):(1)$$TB\;\left(\%\right)=\left(\frac{m_{DA}\times Eq.W_{NCO}}{Eq.W_{DA}\times m_{NCO}}\right)\times 100$$
where *m_DA_* and *m_NCO_* are the mass of the DA adduct and the isocyanate prepolymers in the feed, while *Eq.W_DA_* and *Eq.W_NCO_* are the equivalent weights of the DA adduct (277.3 g/eq. for **DAF** and 232.2 g/eq. for **DAR**) and isocyanate prepolymer (2471 g/eq.), respectively. We assume that all the −OH groups in DA react with −NCO groups in isocyanate prepolymers.

The prepared adhesive films presented the characteristics shown in Figure 2, and it could be clearly noted that a worse dispersion was obtained when **DAR** was used. The films with **DAF** appeared transparent without inhomogeneities, independently of whether FA was used or not. On the contrary, the film with **DAR** prepared without FA presented many bubbles or agglomerations due to the low solubility of **DAR** in the system. In contrast, the film prepared using FA was completely homogeneous with no bubbles present. However, this sample (PUR-B10) contained less than 14% of TB, which may be insufficient for triggering significant changes in the chain mobility of the adhesive at high temperature, despite the occurrence of the rDA reaction. Different polyol:DA:FA ratios were investigated with higher TB contents and the results are shown in Appendix A. From all these results, it can be concluded that the flexible adduct **DAF** disperses more efficiently than **DAR** in the solvent-based system. Moreover, the addition of small amounts of FA improves the dispersion of both DA adducts into the adhesive, but it is also expected to influence the mobility of the polymeric adhesive chains after the rDA reactions, as shown below.

Before evaluating the thermoreversibility of the new polymeric adhesives, the thermal stability and mechanical properties were investigated and compared with commercial PUR. Appendix A shows representative TGA and DTGA curves of different solvent-based adhesives. The degradation behavior of the solvent-based adhesives was very similar, despite the addition of the adducts, to the analogous characteristic degradation temperatures of PUR. The samples exhibited two degradation stages with maximum degradation rates at ca. 315–325 °C and 435–455 °C, respectively, as already observed in similar systems [20]. In these kinds of systems, the first weight loss is assigned to the degradation of the hard segment as a consequence of the relatively low thermal stability of the urethane bonds, and the second process is attributed to soft segment decomposition [21,22]. In this study, a slight stabilization in the first process is observed for all the samples containing the adducts, which seems to be proportional to the adduct content, regardless of the type of adduct, which can be due to a stabilization of the urethane bonds by the DA moieties. Regardless, we can clearly confirm that the incorporation of both adducts did not significantly alter the thermal stability of the PUR.

Appendix A shows storage modulus values, *E’*, for the PUR and for selected modified solvent-based adhesives. For this case, the addition of 13.7 wt.% of TB produces an *E’* improvement that is outside experimental error (*E’* = 12 ± 0.2 MPa for PUR-B01 and *E’* = 14.1 ± 0.3 MPa for PUR-B03). Other studies in similar systems have shown that *E´* increases with the amount of TB [23]. In our work, this *E’*-rise seems to be associated with an enhanced network rigidity as a consequence of the incorporation of DAF in the PUR network. Furthermore, for the same amount of TB, DAR appears to produce an additional mechanical improvement (*E’* = 17.2 ± 0.1) due to the superior rigidity of the DAR moiety.

The thermoreversible performance of the adhesives was investigated, taking into account all the parameters described above with special emphasis on the influence of the chemical structure of the DA adducts and the ratio between the different adhesive components. First, the rDA reaction in the adhesives was studied by Fourier transform infrared spectroscopy. Figure 3 shows the evolution of the ATR-FTIR spectra with three different thermal treatments, as explained in the experimental part, for the sample containing **DAF** and prepared without FA. In particular, this sample contains 60 mol% of OH groups from polyol and 40 mol% from **DAF** (PUR-B04 in Table 2). Particular attention was paid to the region between 700 and 680 cm^−1^ (highlighted with a yellow box), a relatively clear region of the spectrum where the alkene C−H out-of-plane bending vibration of the BMI molecule appeared. In principle, as the rDA/DA cycle takes place, the BMI compound will be produced or consumed and its signal is expected to increase/decrease correspondingly [19,24,25]. As can be observed, at room temperature (black trace) there was no detectable band in this region of the spectrum, which means that the DA adduct structure was preserved in the adhesive system. After thermal treatment at 120 °C for 15 min, a band centered at 690 cm^−1^ appeared (red trace). This confirms that the thermal treatment triggered the rDA reaction and free BMI was obtained. This is a very important result as it confirms that the reaction previously demonstrated for DA adducts in solution, explained above, also occurs when the DA molecules are incorporated into the adhesive system. In addition, for the sample treated at 120 °C and then annealed at 60 °C, the BMI band was reduced significantly, suggesting that the DA closed adduct was formed once again and the adhesive network was partially reconstructed (blue trace). Finally, a subsequent treatment of this last sample at 120 °C also showed the BMI band (green trace), although in this case its intensity appeared to be slightly lower than in the case of the first cycle. However, it is important to highlight that the rDA/DA cycle can, in principle, be repeated at least twice, which is very important for the sustainability and application of the adhesive.

The same behavior was observed for the sample with the same composition but with **DAR** as the thermoreversible moiety (PUR-B11 in Table 2). In this case, the intensity of the BMI signal after the rDA reaction was higher than in the case for **DAF**, with the difference being more pronounced in the first cycle. As both samples had the same composition, this result suggests that the **DAR** moiety is more sensitive to thermal treatment than the **DAF** moiety at the evaluated temperature. Finally, the samples prepared in the presence of FA with **DAF** and **DAR** (PUR-B05 and PUR-B12 in Table 2) showed a similar behavior as previously observed but with the intensity of the BMI signal slightly lower due to the lower content of TB in comparison with the samples prepared without FA.

DSC was also used to monitor the rDA reaction in the modified PUR samples [9,24,25,26,27]. Figure 4 shows the DSC curves of the adhesives modified with **DAF** compared with that without adduct. With the exception of the unmodified PUR (PUR-B01), all samples presented a broad endothermic peak ranging from 90 to 150 °C with a maximum at around 130 °C. This transition correlates with the observations in the ATR-FTIR results and is assigned to the rDA reaction. It can be clearly observed that the enthalpy increased as the TB content increased (from PUR-B03 to PUR-B07 in Figure 4).

The reversibility of the DA reaction, i.e., whether the debonding–bonding process could be repeated over several cycles, was also evaluated by DSC. Figure 5 includes the consecutive heating scans recorded following the cycles defined in the experimental part for the sample with the highest TB content using **DAF** (PUR-B06). A characteristic endothermic peak ranging from 90 °C to 150 °C was observed for each heating cycle, corresponding to the rDA reaction. It can be observed that the enthalpy calculated for each rDA reaction (included in Figure 5) slightly decreased after the first heating scan but then remained constant upon the application of subsequent thermal treatments. These results indicate that after the first rDA reaction, the DA structure is not completely reconstructed during the isothermal treatment at 60 °C [27,28,29], but in the following cycles the DA recovery reaches a steady value. These findings are quite consistent with the ATR-FTIR observations. Finally, it is important to highlight that the rDA/DA cycle can be repeated through multiple cycles; we tested up to nineteen times. This is extremely relevant for the possible reuse of this adhesive or its application for the separation of laminated films.

DSC was also used to evaluate the influence of the chemical structure of the DA adduct on the reversibility of the DA reaction. Figure 6 compares the first heating scans obtained for the samples PUR-B05 and PUR-B12; both samples had a composition of 60 mol% OH groups from polyol, 10 mol% from FA, and 30 mol% from **DAF** or **DAR**, respectively. The results indicate that similar enthalpy values are obtained regardless of the type of adduct used in the preparation of the sample, which is consistent with the fact that both samples had similar TB contents. Moreover, the maximum endothermic peak temperature was similar for both samples in contrast with results reported very recently for similar adducts [9]. In spite of these differences, it should be noted that at the DSC heating rate in our work, the endotherms corresponding to exo and endo stereoisomers overlapped and the chemical structures of the polyurethanes employed in both articles were significantly different. Regarding the glass transition (*T_g_*) of the adhesives, a clear influence of DA content was observed and will be described elsewhere, together with the mechanical properties.

A comparison of adhesive cyclability as a function of the different DA adducts was also conducted. Figure 7 shows the variation in the rDA enthalpy obtained from DSC for adhesives with both adducts from 20 DA/rDA cycles. It can be seen that both presented a similar trend: an initial drop in the enthalpy and certain stability during subsequent cycles. It can also be pointed out that, for the adhesives with **DAF,** the enthalpy stabilized at values higher than those for **DAR**, which means that the network reconstruction during the annealing at 60 °C was higher in this case. This can be attributed to a higher mobility of the chains containing the flexible adduct that facilitates the recombination of the reactive groups in the adhesives to regenerate the DA bond, which is corroborated by the ATR-FTIR data (Figure 3).

It should not be forgotten that our main objective was to introduce an easy-to-split functionality into the PU adhesives to allow their possible reuse and to facilitate the separation of multilayer films, whilst maintaining their characteristic properties during regular use. The use of DA reactions in polyurethanes has been reported in the case of self-healing materials, whereby an increase in temperature provoked an increased fluidity of the sample due to a reduction in the molecular weight of the polymer [10]. Therefore, once it was demonstrated that rDA/DA takes place in the adhesives, the influence of the rDA reaction on their physical properties was investigated by analyzing the temperature-induced changes in the fluidity of the adhesive using the procedure described in the last paragraph of the experimental section.

For samples with 70 mol% OH groups originating from polyol and 30 mol% from the DA adducts (PUR-B02 and PUR-B09), although the retro-DA took place, no effect on the physical properties of the adhesives was seen. The addition of 10 mol% of FA (PUR-B03 and PUR-B10) did not have an effect on the physical properties either. This was due to the fact that the TB content was below a critical value, and an increased adduct concentration was required. However, the amount of adduct was limited by its solubility in the adhesive. 

The upper half of Figure 8 presents images of the samples containing 60 mol% of OH moieties from polyol and 40 mol% from the DA adducts (without FA) after thermal treatment. In this case, it was observed that only the sample containing **DAR** could be slightly reshaped in the central region of the specimen (marked in red box). Nonetheless, the FTIR results confirmed that the retro-DA reaction had indeed taken place in both samples, suggesting once again that the TB content was still insufficient to modify the physical properties of the adhesive. Considering the composition of these samples, it was estimated that a TB content above around 27% was needed.

Subsequently, the addition of FA to improve the dispersion of the DA moieties was evaluated. Samples containing 60 mol% of OH from polyol, 30 mol% of **DAF** (PUR-B05) or **DAR** (PUR-B12) and 10 mol% of FA were employed, and the results are presented in the lower half of Figure 8. It can be clearly observed that when FA was used both formulations demonstrated flow at a high temperature with reshaping of the central region of the specimens. Therefore, the addition of a small amount of FA significantly improved chain mobility after the rDA reaction, enabling the flow and reshaping of the PUR, regardless of the type of DA adduct used. Furthermore, this behavior was observed for lower TB content. It should be noted that FA can be obtained from natural sources and that any excess of this solvent observed after DA synthesis, rather than presenting a problem, is beneficial for the performance of the adhesives.

From the results described for the solvent-based PUR, it was concluded that a minimum TB content must be surpassed for the rDA reaction to have a significant influence on the physical properties of the adhesive. Moreover, the addition of specific amounts of FA to the PUR formulation significantly improved the flow and reshaping performance of PUR at high temperatures, as well as reduced the critical TB content required for flow. Thus, FA (used as a reagent and solvent for the synthesis of the DA molecules) plays a dual role as it also assists with the incorporation of the adducts, especially **DAR**, in the adhesive systems as described above, and can be employed as a tool for tuning the flow performance of the adhesives by varying the formulation.

### 3.3. Incorporation of Thermoreversible Molecules into Solvent-Free PUR Adhesives

A similar study to that described for the solvent-based system in Section 2 was conducted for a PUR solvent-free system. In this case, the incorporation of adducts was more challenging as no solvent was used to aid its dispersion into the adhesive formulation. However, the objective was the same as in the case of the previous adhesive system, i.e., obtaining new adhesives with the same bonding capacity as the commercial ones, with the possibility to be easily detached after use, facilitating the delamination of multilayer systems and their recycling.

Commercial solvent-free NCO-terminated and OH-terminated components were used. Regarding the adducts, only **DAF** was investigated due to the very low solubility of **DAR** without the assistance of an appropriate solvent. The incorporation of the adduct into the solvent-free adhesive required much longer mixing times via a two-step procedure, as described in the experimental part. Adhesives with different polyol/DA adduct/FA ratios were prepared and are listed in Table 3. As for the solvent-based system, in this case, the TB values were calculated from Equation (1), where *Eq.W_NCO_* takes a value of 290 g/eq. We assumed that all the −OH groups in DA reacted with the −NCO groups in isocyanate prepolymers. Adhesive samples with additional FA were prepared for comparison. A good dispersion of DAF was observed for all compositions (Appendix A in ESI), and it was found that there was no need to use FA to improve it. Nonetheless, samples prepared with FA were also studied considering the positive effect that FA played in the reshaping/fluidity of the solvent-based adhesives.

The thermal stability of the new solvent-free adhesives was evaluated and the results are shown in Appendix A. As for the solvent-based adhesives, the TGA of the modified adhesives showed the same shape as the commercial PUR with the two processes described previously. A slight stabilization was perceived for samples with DAF as well, but what it is important here is that the new adhesives displayed similar thermal stability to the unmodified PUR. 

The mechanical properties for this system were also tested by nanoindentation, and quite a remarkable *E’*-increase was found with the incorporation of 15 wt.% of TB (see Appendix A). This could be related to the relatively lower free volume of the solvent-free adhesives with respect to the solvent-assisted one. Reduced free volume can be expected to hinder chain mobility and enhance the material’s elastic response. Indeed, the glass transition temperatures of the solvent-free adhesives were significantly shifted to higher temperatures compared to those of solvent-based adhesives, approaching room temperature values in agreement with reduced chain mobility. These are very interesting results and a full study on the viscoelastic properties of these adhesives by indentation testing and parallel-plate rheometry is currently underway and will be reported soon.

As in the solvent-based systems, the occurrence of the rDA reaction in the solvent-free adhesives was investigated by ATR-FTIR and DSC. Figure 9A shows the evolution of the ATR-FTIR spectra under different thermal treatments for the sample containing **DAF** (without FA) and 12 wt.% of TB (PURSF-04). At room temperature (black trace), no bands were seen in the regions of the spectrum where the BMI should have appeared. However, after heating at 120 °C for 15 min (red trace), three new signals clearly appeared (yellow boxes) assigned to the BMI compound, which corroborates the opening of the dynamic covalent bonds. The application of the isothermal treatment at 60 °C for 20 h (blue trace) significantly decreased the intensity of the BMI peaks, but the peak at 690 cm^−1^ was still noticeable, indicating a partial DA network reconstruction. The bands assigned to BMI were observed after successive thermal treatments confirming rDA/DA repeatability. The behavior of the samples prepared with FA is shown in Figure 9B, and a similar trend can be observed regarding the repeatability of the rDA/DA cycles. However, in this case, after the isothermal treatment at 60 °C, the network reconstruction led to spectra (blue and green traces) that were similar to the starting one (black trace) and no remaining BMI signal could be distinguished. This suggests that the reconstruction of the DA network appears to be more favorable when FA is incorporated into the formulation.

The study of the rDA reaction by DSC demonstrated the presence of this transition in all the modified solvent-free adhesives with an increase in the enthalpy associated with the process as the TB content increased (see Appendix A in ESI). The rDA/DA reversibility in the solvent-free system was also measured by DSC. Figure 10 presents the consecutive heating scans recorded for a sample containing 9 wt.% of TB and with FA (PURSF-07). As expected, an endothermic peak ranging from 90 °C to 150 °C corresponding to the rDA reaction was observed for each heating cycle. The rDA process was clearly observed at least up to the 10th cycle. However, in contrast to the solvent-based system, the rDA enthalpy progressively decreased with the application of successive heating cycles, indicating a gradual loss in the DA network reconstruction efficiency (Figure 11). This is supported by the changes in the glass transition temperature (*T_g_*). From Figure 10, it can clearly be seen that the *T_g_* decreased with the number of rDA/DA cycles due to the increasing mobility of the system.

Finally, the influence of TB content on the fluidity of the adhesives was analyzed. Figure 12 (upper half) shows the results for samples prepared without FA. It can be observed that none of the adhesives, not even that with the highest thermoreversible bond content, presented any reshaping after thermal treatment. Despite the presence of **DAF** in its structure, these adhesives were not able to flow after the thermal treatment. However, samples where **DAF** was combined with FA showed a very different behavior (Figure 12, lower half). In this case, the adhesives were able to flow and reshape at 120 °C, even at the lowest **DAF** content. It is worth noting that the shape of the control sample, which included FA in its formulation (PURSF-02), remained unaltered regardless of the thermal treatment. Therefore, it can be concluded that in this solvent-free system the combination of FA and **DAF** is important to produce materials with the fluidity required for their potential application in delaminating multilayer systems after thermal treatment.

This manuscript is aimed at the preparation of adhesives that after use can be modified by heating, losing their adhesion and facilitating the recycling of multilayer polymer laminates. Thus, in order to verify that adequate adhesion is maintained when applying the new adhesives, lab-laminated low-density polyethylene (LDPE) and polyethylene terephthalate (PET) sheets of different thicknesses were prepared using a commercial solvent-free PUR with its corresponding **DAF** adduct-modified adhesive (see ESI for details) and t-peel tests were undertaken to assess their peel resistance. The initial adhesion of the laminates with **DAF**-modified adhesive was found to be similar to that of the commercial adhesives, reaching the required values of around 2.5 N/25 mm. Samples of the laminates were then immersed in boiling water for 30 min as an initial trial method for the activation of the rDA reaction. The values of peel resistance of the tested laminates after the thermal treatment are provided in ESI Appendix A. It was found that after the thermal treatment in boiling water the peel resistance fell significantly in laminates with the **DAF**-modified adhesive, whereas for the commercial formulation, the reduction was markedly inferior. These preliminary results confirm our initial hypothesis and demonstrate the potential of thermoreversible Diels–Alder bonds for the delamination of multilayer systems for packaging. An in-depth study of the adhesion properties of pilot-scale laminated systems and their variation as a function of thermal treatment and the optimization of the heating strategy employed for delamination is currently underway and the results will be reported at a later date.

## 4. Conclusions

New adhesives have been developed with the main objective of incorporating functionalities that allow them to be reusable, to be employed in delamination and facilitate the recycling of multilayer systems for packaging. For this purpose, two DA adducts of different natures (**DAF** and **DAR**) with dynamic covalent bonds were incorporated into commercial solvent-based and solvent-free polyurethane adhesives.

Differences in solubility, dispersion and sensitivity to thermal treatment were related to the chemical structure of the adduct. In particular, **DAR** is only soluble in solvent-based adhesives and cannot be incorporated into the solvent-free system. DSC and ATR-FTIR experiments demonstrated the existence of the retro-DA reaction and its reversibility regardless the type of adduct and adhesive. The bonding/debonding process, i.e., the rDA/DA cycle, can be repeated up to ~20 times in the solvent-based adhesive for both adducts, although the drop in enthalpy in the first cycles is higher in the case of **DAR**, suggesting a lesser extent of network reconstruction in this case. In contrast, for the solvent-free system with **DAF**, a gradual loss in the DA network reconstruction efficiency was observed with the rDA/DA cycles.

The physical properties of the adhesives can be tailored by regulating the feed composition (especially the ratio between different diols) and the final concentration of DA bonds. A minimum content of thermoreversible bonds needs to be surpassed for the rDA reaction to have a significant influence on the physical properties of the adhesive. However, the addition of furfuryl alcohol to both solvent-based and solvent-free systems has been demonstrated to improve the flow and reshaping performance of the polyurethane adhesives at high temperatures, and can be used as a formulation tool to control adduct concentration since good flow properties can be achieved with lower TB content.

The new functionalities incorporated in the adhesives provide the possibility to lose adhesion on demand, allowing the post-consumer separation of the packaging components and their reuse. The new adhesives represent a contribution towards a circular economy in plastics by offering alternative strategies for the recycling of multilayer packaging systems. In addition, we envisage that the strategy employed to develop these new reusable adhesives can be applied to other adhesive applications.

## 5. Patents

H.J. Salavagione, H. Seyler, S. Quiles, P. Shuttleworth, G. Ellis, Marián A. Gómez-Fatou. Reversible adhesive material for applications in packaging recycling. REF#: EP21383023. Priority date: 12 November 2021. Holder entity: CSIC.

## Figures and Tables

**Figure 1 polymers-14-03402-f001:**
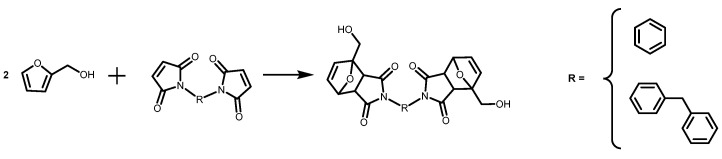
Diels–Alder reaction of furfuryl alcohol (FA) with two bismaleimides to yield two DA adducts (**DAF** and **DAR**).

**Figure 2 polymers-14-03402-f002:**
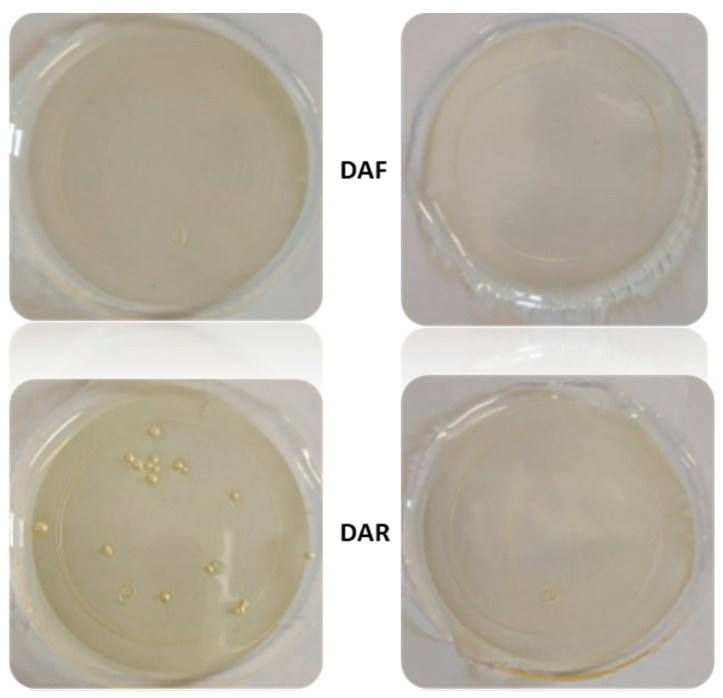
Comparison of adhesive films with 70 mol% −OH groups from polyol without FA (**left**) and with FA (**right**) for formulations incorporating DAF (PUR-B02, 18.7% TB and PUR-B03, 13.7% TB) or DAR (PUR-B09, 19.5% TB and PUR-B10, 13.7% TB).

**Figure 3 polymers-14-03402-f003:**
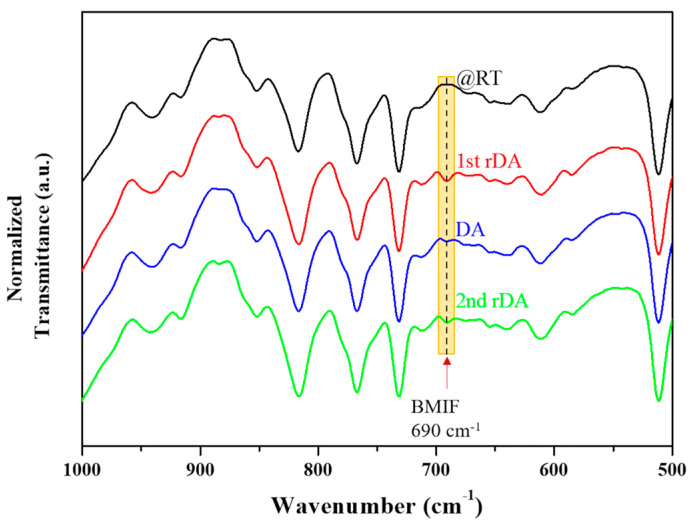
ATR-FTIR spectra showing the evolution of the flexible BMI (BMIF) signal at different thermal treatments for the sample PUR-B04, 25.2% TB. The band marked BMIF at 690 cm^−1^ is a C=C-H deformation mode characteristic of the bismaelimide.

**Figure 4 polymers-14-03402-f004:**
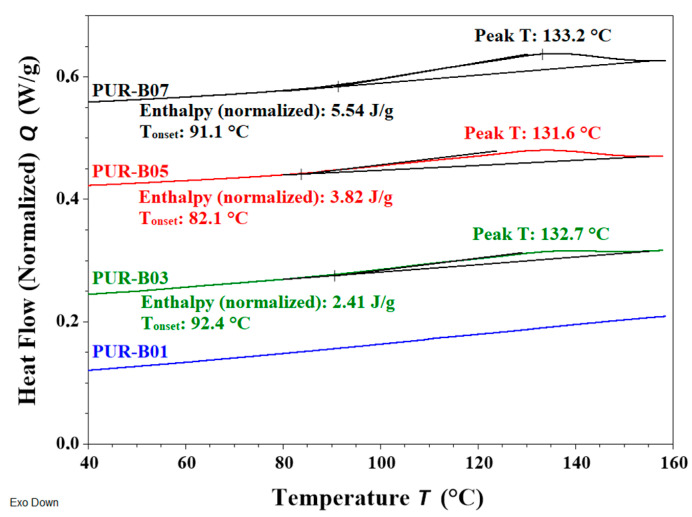
DSC heating curves at 10 °C/min for PUR-B01 and those samples prepared with DAF/FA: PUR-B03 (13.7 %TB), PUR-B05 (18.8 %TB) and PUR-B07 (25.7 %TB).

**Figure 5 polymers-14-03402-f005:**
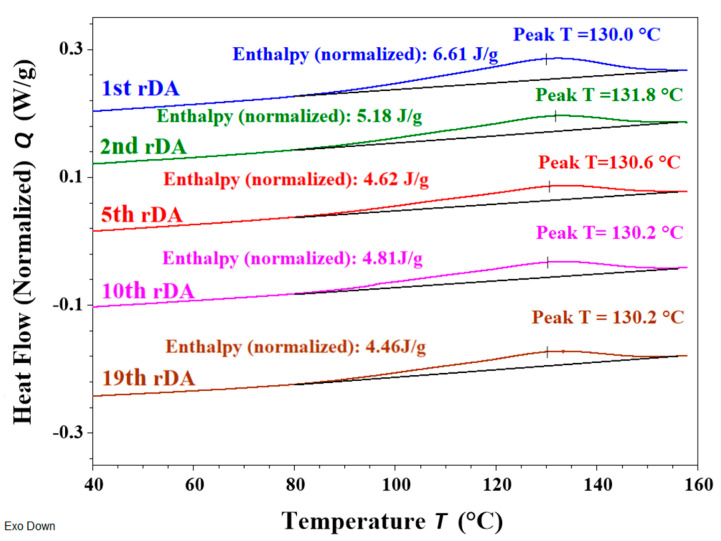
DSC consecutive heating scans at 10 °C/min for the sample PUR-B06 (33%TB).

**Figure 6 polymers-14-03402-f006:**
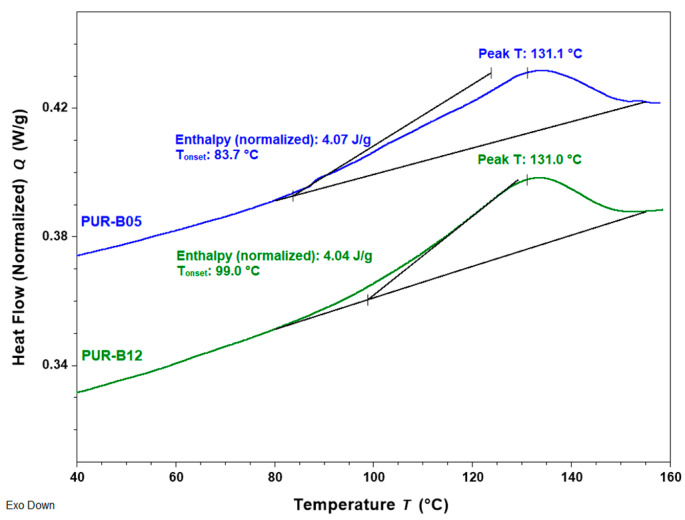
DSC heating curves at 10°C/min for the samples PUR-B05, 18.8% TB (with DAF) and PUR-B12, 19.9% TB (with DAR).

**Figure 7 polymers-14-03402-f007:**
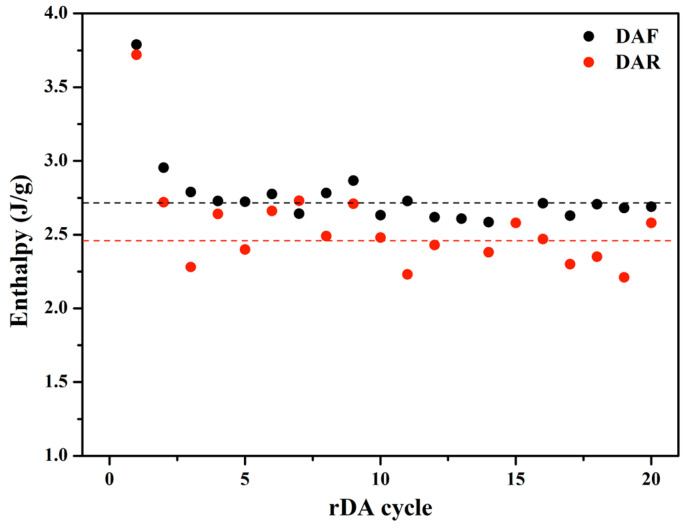
Variation in the enthalpy of the rDA reaction of solvent-based PUR adhesive containing DAF (PUR-B05, 18.8%TB) and DAR (PUR-B12, 19.9%TB) with the number of DA/rDA cycles.

**Figure 8 polymers-14-03402-f008:**
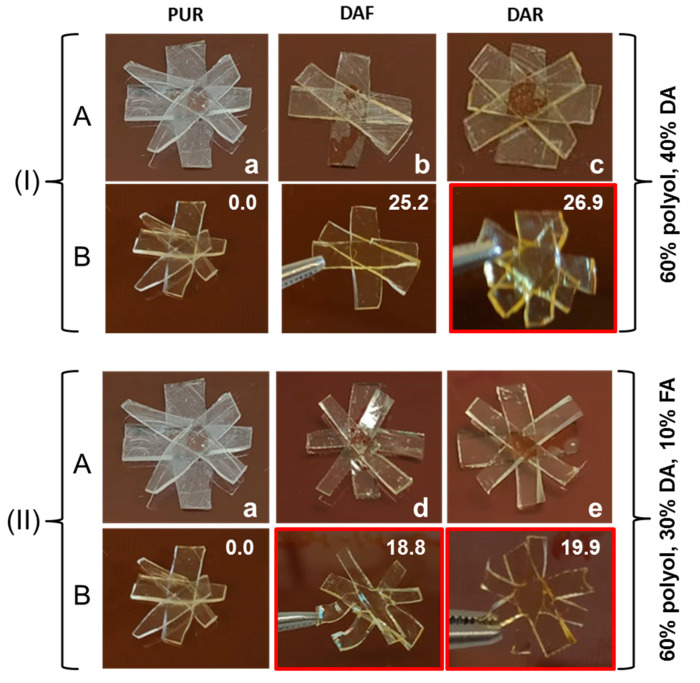
Photographs of adhesive samples before (**A**) and after (**B**) thermal treatment at 120 °C for 15 min. (I): (**a**) PUR-B01 reference, (**b**) PUR-B04 (25.2%TB) and (**c**) PUR-B11(26.9%TB) samples. (II): (**a**) PUR-B01 reference, (**d**) PUR-B05 (18.8%TB) and (**e**) PUR-B12 (19.9%TB) samples.

**Figure 9 polymers-14-03402-f009:**
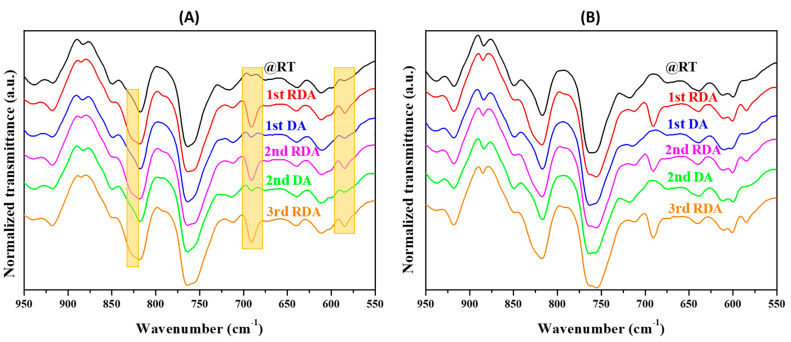
ATR-FTIR spectra showing the evolution of the flexible BMI signal at different thermal treatments for the samples PURSF-04 (**A**) and PURSF-08 (**B**) with 12%TB. The orange boxes correspond to regions where deformation modes characteristic of BMIF appear.

**Figure 10 polymers-14-03402-f010:**
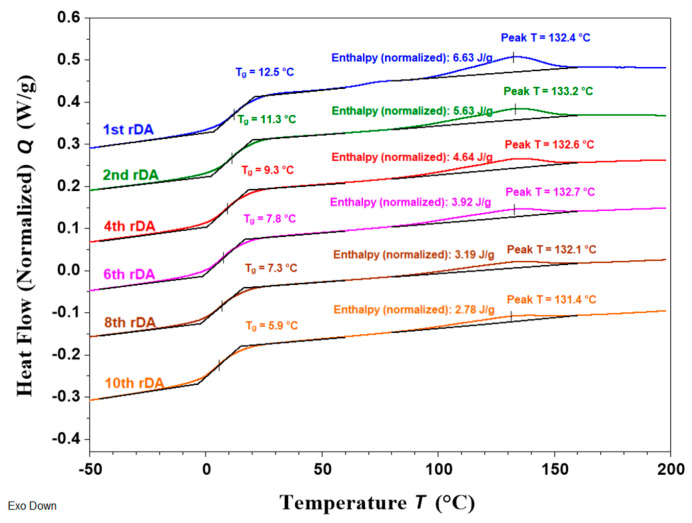
DSC consecutive heating scans at 10 °C/min for the sample PURSF-07 (9%TB).

**Figure 11 polymers-14-03402-f011:**
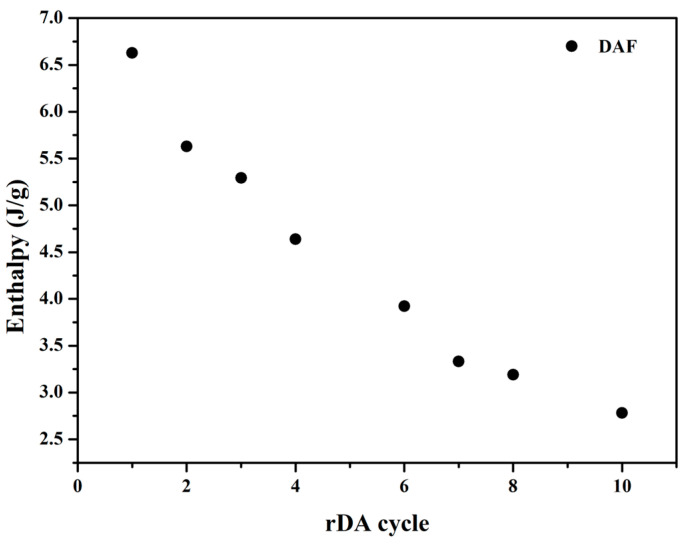
Variation in the enthalpy of the rDA reaction of solvent-free PUR adhesive containing DAF (PURSF-07, 9%TB) with the number of DA/rDA cycles.

**Figure 12 polymers-14-03402-f012:**
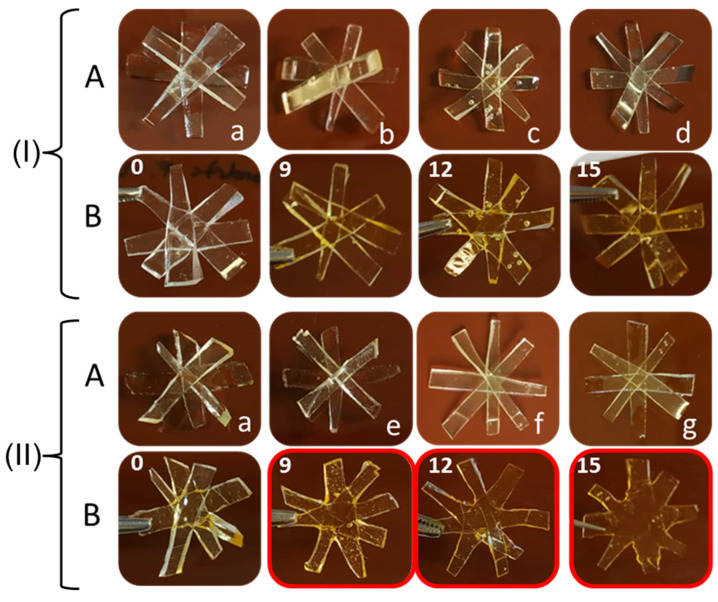
Photographs of adhesive samples before (**A**) and after (**B**) thermal treatment at 120 °C for 15 min. (I): (**a**) PURSF-01 reference, (**b**) PURSF-03 (9 %TB), (**c**) PURSF-04 (12 %TB) and (**d**) PURSF-05 (15 %TB) samples. (II): (**a**) PURSF-01 reference, (**e**) PURSF-07 (9 %TB), (**f**) PURSF-08 (12 %TB) and (**g**) PURSF-09 (15 %TB) samples.

**Table 1 polymers-14-03402-t001:** Experimental conditions optimized for the synthesis of the DA molecules in furfuryl alcohol.

Name	FA:BMI Ratio	T/°C	Time/h	DA Purity	Yield
DAF	6.5:1	40	24	96%	60%
DAR	14:1	55	21	93%	72%

**Table 2 polymers-14-03402-t002:** Solvent-based adhesives’ nomenclature and composition.

Name	NCO/OH Mole Ratio	Type of DA	% OH from POH	% OH from DA	% OH from FA	DA Content wt.%	Mass DA Adduct/mg	% TB *
PUR-B01	1.5	-	100	0	0	-	0	0
PUR-B02	1.5	DAF	70	30	0	2.9	21.0	18.7
PUR-B03	1.5	DAF	70	20	10	2.1	15.4	13.7
PUR-B04	1.5	DAF	60	40	0	4.0	28.3	25.2
PUR-B05	1.5	DAF	60	30	10	3.0	21.1	18.8
PUR-B06	1.5	DAF	50	50	0	5.2	37.0	33.0
PUR-B07	1.5	DAF	50	40	10	4.1	28.8	25.7
PUR-B08	1.5	DAF	50	25	25	2.6	18.2	16.2
PUR-B09	1.5	DAR	70	30	0	2.5	18.3	19.5
PUR-B10	1.5	DAR	70	20	10	1.9	12.9	13.7
PUR-B11	1.5	DAR	60	40	0	3.5	25.3	26.9
PUR-B12	1.5	DAR	60	30	10	2.6	18.7	19.9
PUR-B13	1.5	DAR	50	50	0	4.5	31.9	33.9
PUR-B14	1.5	DAR	50	40	10	3.3	23.0	24.5
PUR-B15	1.5	DAR	50	35	15	3.2	22.1	23.5
PUR-B16	1.5	DAR	50	30	20	2.7	19.1	20.3
PUR-B17	1.5	DAR	50	25	25	1.9	13.1	13.9

* Assuming that all the incorporated DA adduct has reacted with the NCO groups.

**Table 3 polymers-14-03402-t003:** Solvent-free adhesives’ nomenclature and composition.

Name	NCO/OH Mole Ratio	Type of DA	% OH from POH	% OH from DA	% OH from FA	DA Content wt.%	Mass DA Adduct/mg	% TB
PURSF-01	1.3	-	100	0	0	0	0	0
PURSF-02	1.3	-	80	0	20	0	0	0
PURSF-03	1.3	DAF	88	12	0	4.8	86.1	9
PURSF-04	1.3	DAF	84	16	0	6.5	114.7	12
PURSF-05	1.3	DAF	80	20	0	8.1	143.4	15
PURSF-06	1.3	DAF	76	24	0	9.7	172.1	18
PURSF-07	1.3	DAF	76	12	12	5.0	86.1	9
PURSF-08	1.3	DAF	68	16	16	6.8	114.8	12
PURSF-09	1.3	DAF	60	20	20	8.6	143.5	15

## Data Availability

Not applicable.

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
