# Peer review of "Designing New Sustainable Polyurethane Adhesives: Influence of the Nature and Content of Diels–Alder Adducts on Their Thermoreversible Behavior"

_polymers, 2022, doi:10.3390/polym14163402_

Round 1

Reviewer 1 Report

This manuscript focus more on chemistry instead of adhesive materials. More specifically, why DA reaction should be incorporated in PU as adhesives? It helped the adhesion, recycling, or self-healing? The aim must be clarified and experiments must be performed to prove the advantages. 

Why disulfide bonds cannot be used instead? 

Lots of useful information, for example, sample name table, calculation of TB, transparency, should be moved to main text.

Finally, the properties, such as thermal stability, mechanical properties of PU adhesives must be added. Otherwise, as I said, the manuscript focus a lot on chemistry.

some references could be helpful

Macromolecular Materials and Engineering 305 (1), 1900578 2020

Progress in Organic Coatings 147, 105876 2021

Progress in Organic Coatings 133, 357-367 2019

Author Response

see attachement

Reviewer 2 Report

In this manuscript, the authors investigated the effect of two Diels-Alder (DA) adducts as a third component in the curing process of solvent-based and solvent-free polyurethanes, bonding/debonding properties of the adducts were also studied. This research is interesting and meaningful, some comments should be considered as follow:

1.     Besides NMR, it could be better to use mass spectrometry to identify the structure of two DA adducts.

2.     In Table 1, the authors only screened two conditions of FA: BMI ratio, temperature and reaction time for the synthesis of the DA molecules, and the optimized yield was moderate (72%). Is there any attempt to improve the yield?

3.     What methods were used to determine the purity and yields of the DA molecules in Table 1?

4.     Is there any microscopic structure of the adhesives? Such as SEM, TEM?

5.     In Figure 3, the signal highlighted with a yellow box is too weak to identify as C-H bending vibration, and the IR peaks should be labeled.

Author Response

see attachement

Reviewer 3 Report

The manuscript fitting in the modern field of self healing materials and the application of dynamic bonds. However, already many examples of diels-alder networks are published. Furthermore, the polymer backbone a polyol is not described in detail. It is necessary to have much more information about it. Therefore, it is not possible to proof the given content values as for example the 60 mol% of OH. In the introduction it is written that the adhesive will be used for adhesion of multilayer packaging. However, not even one experiment is shown in which the adhesion properties have been measured. Therefore, the introduction and the conclusion do not fit together. Furthermore, the literature 14 is not published or even accepted. The article cannot be accepted in this form. 

Author Response

see attachement

Round 2

Reviewer 3 Report

The article is improved but has still weak points. It should be mentioned how much of each monomer (crosslinker) or solutions was in the feed of the reaction. The C=C-H vibration can be found in the maleimide and as well in the diels alder adduct.  I found a mistake in the novel reference 14 in the reference table (it is from 1993 and not from 1983). I very much appreciate the adhesion force  experiment. However, why the PE film was not activated before adhesion. It is well know that pristine PE show a very weak adhesion. Therefore, PE is commonly oxidized on its surface before adhesion as for example with treatment of KMnO4, plasma or corona treatment. I am missing absolute values for the adhesion. I am still missing an experiment which shows that the sample loses its adhesion after heating up to the retro-diels alder reaction as promised in the introduction. I mean in sense of to measure the peel force after curing at 50 or 90°C and than another time after heating the temperature to the expected diels-alder reaction.   

Round 3

Reviewer 3 Report

The manuscript have been very much improved. However, the calculation of TB is an remaining question. When I try to recalculate than I calculate mPolymer + mDA, because in the experiment you take 1 gram of polymer and then you add the molecule which forms the Diels-Alder adduct. From the sum of this I can calculate the amount of mDA which I need in the equation for TB. The amount the polymer is given by 1 g. The values for TB which I calculate differ from your values. You should make the equation more clear or simply to add a table which every sample and the feeds you have used. The only open point is the TB values and how you calculate them. Where I can find mDA? Or should I calculate it from the weight% of DA. I did it like this and found different values.   

Author Response

Clearly the mass of the adducts (mDA) facilitates the calculation, as the reviewer points out, and we have added the amount of each adduct employed for all the samples with extra columns in Tables 2 and 3.